# Progress of Microencapsulated Phycocyanin in Food and Pharma Industries: A Review

**DOI:** 10.3390/molecules27185854

**Published:** 2022-09-09

**Authors:** Yang Li, Xu Li, Zi-Peng Liang, Xin-Ying Chang, Fu-Tong Li, Xue-Qing Wang, Xi-Jun Lian

**Affiliations:** Tianjin Key Laboratory of Food Biotechnology, College of Biotechnology and Food Science, Tianjin University of Commerce, Tianjin 300134, China

**Keywords:** phycocyanin, microencapsulation, biologically active substance, stability, applications

## Abstract

Phycocyanin is a blue fluorescent protein with multi-bioactive functions. However, the multi-bioactivities and spectral stability of phycocyanin are susceptible to external environmental conditions, which limit its wide application. Here, the structure, properties, and biological activity of phycocyanin were discussed. This review highlights the significance of the microcapsules’ wall materials which commonly protect phycocyanin from environmental interference and summarizes the current preparation principles and characteristics of microcapsules in food and pharma industries, including spray drying, electrospinning, electrospraying, liposome delivery, sharp-hole coagulation baths, and ion gelation. Moreover, the major technical challenge and corresponding countermeasures of phycocyanin microencapsulation are also appraised, providing insights for the broader application of phycocyanin.

## 1. Introduction

Phycocyanin (PC) separated from *Spirulina platensis* is a soluble pigment-protein which is formed from the thioether covalent bond between open-chain tetrapyrrole chromophores (called Phycocyanobilin (PCB) or Phycoerthrobilin (PEB)) and apoproteins (protein components which conjugate the chromophores). As a food additive, it is permitted by many countries in the world [1], and its global marketing value is expected to reach USD 245.5 million in 2027 [2]. PC has anti-inflammatory, -oxidant, -tumor, and -modulating immunity and other biological functions [3] and been commonly used as an additive in food or cosmetics. By folding the primary structure of apoprotein, the compact secondary, tertiary, or quaternary structures are formed; meanwhile, the linear chromophores are rearranged into a cyclic form, and the cyclic chromophores as well as the conjugated apoprotein form a protein–chromophore complex. Moreover, the complex increases the wavelength utilized by *Spirulina platensis* for photosynthesis, extends the photosynthesis’ wavelength range, and gives the blue color of PC [4]. Evidence has showed that the antioxidant ability of PC was higher than that of its apoproteins, which indicated that the antioxidant ability of PC was mainly provided by the chromophore [5]. Nicotinamide adenine dinucleotide phosphate (NAPDH) is a coenzyme, also called reduced coenzyme II, which is formed in the light reaction stage of photosynthesis. It enters the carbon reaction with ATP and participates in the fixation of CO_2_. The tetrapyrrole structure of PC chromophore could activate NADPH oxidase through hydrophobic interaction and inhibit the production of NADPH-dependent superoxide [6]. PC chromophore also played an anti-inflammatory role in inhibiting nuclear factor-kappa B (NF-κB) activation in various inflammatory models of animal and cell [7,8], thereby effectively improving renal injury [9] and intestinal inflammation [10]. PC scavenged external free radicals by radical addition reaction at conjugated double bonds of the tetrapyrrole structure’s chromophores, leading to the oxidative degradation of covalently linked chromophores to mesobiliviolin and biliviolins [11] and lost the color of the chromophore itself [12]. In addition, the reasons for PC chromophores easily losing color also include being susceptible to pH, light, temperature, and other environmental factors [13]. Therefore, the color stability and bioactivity of PC are closely related to the existence environmental conditions of chromophores. When PC was either processed or taken orally, the color changed, as did the biological activity, because the former was subjected to external hostile environments, while the latter was subjected to the internal gastrointestinal extremely environment, and then the application range of PC was limited. To overcome the abovementioned shortcomings, targeted delivery protective measures should be given to escape the harmful adversity and to improve the color and structural stability of PC.

Microencapsulation is a micro-packaging technology that encapsulates trace substances with polymer films. Protected substances, called core materials, which can exist in solid, liquid, and gas phase, are wrapped by polymer film-forming materials, known as wall materials [14]. At present, this technology has been successfully applied to encapsulate natural pigments such as anthocyanins [15], curcumin [16], and other plant extracts [17,18], for protection and utilization of such compounds.

Up to now, multifarious strategies have been successfully investigated for the microencapsulation of PC. Table 1 presents the summary of the advantages and disadvantages of the recent encapsulation strategies [15]. Encapsulation of PC by microcapsule technology can significantly enhance its stability and improve its bioavailability both in vivo and in vitro. The prepared microcapsules exhibit the features in the encapsulation efficiency, particle size, and controlled release ability, meeting the need of corresponding fields. Moreover, microcapsules can synergize with the wall material to exert biological activity. This paper reviewed the structure, properties, and physiological functions of PC, summarized the microcapsule technology for PC protection in recent years, and provided a reference for the research and development of PC in different fields.

## 2. The Structure, Properties, and Functions of PC

PC is a blue-binding protein isolated from *Spirulina platensis*, which is an important component of the photosynthetic pigment complex of cyanobacteria and red algae. PC is divided into two types of C- and R-PC. C-PC is mainly derived from cyanobacteria, while R-PC is from red algae, and the major difference between them is the ligands at 155 sites of the β chain, shown in Figure 1. The three ligands of C-PC are all PCB with an open-chain tetrapyrrole structure, while three of the R-PCs are PEB, the isomers of PCB.

PC is usually a mixture of monomer, trimer, and hexamer in the solution. PC monomer is composed of α and β subunits, in which α subunit contains 162 amino acids and β subunit contains 172 amino acids [49]. The chromophores are covalently bound to the Cys84 of the α subunit and to the Cys82 and Cys153 of the β subunit, named α84, β84, and β155, respectively. Due to the different numbers of charged residues on the polypeptide chain, the local structures of chromophores are various [50]. The α or β chains of PC are composed of nine helixes (X, Y, A, B, E, F’, F, G, and H) and form α and β subunits, respectively. Through electrostatic interaction between α and β subunits, monomer of αβ is formed, as shown in Figure 2a. The helical X and Y in α and β subunits formed partially overlapped and crescent-shaped monomer (αβ) by hydrophobic and ion–dipole interaction, and three monomers of αβ form a disc-shaped trimer (αβ)_3_ with central cavity around the central axis, as shown in Figure 2b. Regarding the interaction between α subunits, two (αβ)_3_ trimers with combinations of face-to-face form monoclinic PC hexamer (αβ)_6_, as shown in Figure 2c. The molecular structure and spatial conformation of PC affords it its own special spectroscopic properties.

PC with a bright blue color and bioactivities can be used as a natural pigment or active substance in many fields [51]; however, its instability and easy fading limits its broad applications [52]. The representations of PC polymers and strength of the aggregation force among them significantly depend on the environmental conditions [53]. High temperature and extreme pH values will destroy the protein–chromophore interaction, leading to damage of the structure and state of the chromophore, and eventually resulting in a color change [54]. The PC in aqueous solution is extremely unstable under the light and the heat. When the temperature exceeded 47 °C, PC was degraded, and then the color change occurred [55]. After being exposed to the light intensity at 3 × 10^5^ lux for 24 h in the aqueous solutions with pH = 5 and 7, 80% of PC was degraded [56]. The fading and fluorescence quenching of PC is normally caused by the hydrolysis of PCB, and extreme environmental conditions will seriously destroy its conjugated system. In view of the above situations, if the external compound is encapsulated onto the surface of PC and forms a hydrophobic barrier around PCB, it can stabilize both the color and the fluorescence of PC in an aqueous system and protect PC from degradation.

PC has a wide range of biological functions. In regulating intestinal flora, PC adjuvant therapy could reduce intestinal permeability and increase intestinal barrier function [57]. PC also acted on the site of damaged liver, reduced oxidative stress, and restored liver enzymes and liver cell structure [58]. Increasing studies have showed that PC could inhibit the proliferation of cancer cells by inducing abnormal apoptosis to achieve the anti-cancer effect [59]. Imai et al. [60] found that, when PC was digested by neutral proteases and then taken orally, normal gene expression in the body was maintained or restored, and cognitive impairment caused by Alzheimer’s disease was improved. PC also has a high fluorescence quantum yield. Its porphyrin ring could present affinity with tumor cells and activated oxygen molecules to form singlet oxygen by absorbing photon energy to kill tumor cells [61]. Based on these biological activities of PC, the question of how to develop it into food, medicine, and healthcare products to realize its functions has become a research hotspot in related fields.

At present, the wall materials of microencapsulated PC include natural materials and synthetic materials. Microencapsulation methods of PC involved spray drying, electrospinning, electrospraying, liposome delivery, sharp-hole coagulation baths, and ion gelation. The above materials and methods used for preparing microcapsules effectively solve the problem of poor color and fluorescence stability of PC and improve its bioavailability.

## 3. Microencapsulated PC Wall Materials

PC compounds do not fare well without proper protection measures because they are unstable and sensitive to the environment. When the PC, through oral treatment, is in the stomach, its polypeptide chain will be protonated by gastric acid and then opened by the folded polypeptides and lost in the chromophores’ spatial structure, as will the biological activity. Based on this, when PC is applied in the food and health care industry, the wall material over PC microcapsules must be able to tolerate the extremely acidic environment in the stomach; then, the PC is released again in the intestinal environment. Moreover, PC in the intestinal gives the chromophore full functions on the biological activity. Studies have showed that PC increased the proliferation of intestinal beneficial bacteria, regulated the balance of intestinal microflora, reduced the level of lipopolysaccharide, inhibited the activation of toll-like receptor 4 (TLR4)/myeloid differentiation factor 88 (Myd88)/NF-κB pathway, and downregulated the levels of inflammatory cytokines such as tumor necrosis factor α and interleukin 6 [62,63]. Therefore, the oral delivery of PC could be achieved by selecting proper wall materials which are stable in acidic conditions and dissolve in an alkaline environment.

### 3.1. Natural Polymer Materials

Natural polysaccharides, such as starch, maltodextrin, pectin, etc., can be used as wall materials in the encapsulation process of PC. Typically, PC combined with the natural polysaccharides through some of their specific groups and wrapped inside the formed cage structure by intermolecular forces. The encapsulation process ensured PC stability and biological activity promotion, which has resulted in its wide use as food and pharmaceutical ingredients [64]. In addition, PC could be utilized as a substitute for animal-derived serum. Microencapsulation of PC with polysaccharides could be applied in the food industry [65].

Starch is a high molecular polysaccharide formed by intermolecular dehydration of glucose, which can be divided into amylose and amylopectin. Due to their spiral structure and the formation of cavities at the inner helix center, starch can be employed to encapsulate a variety of bioactive substances. Some kinds of starch can be used as microcapsule wall materials through the retrograde process to form a microcrystal bundle that wraps the compounds. The retrogradation starch, a type of resistant starch used for PC encapsulation, can effectively improve the PC chemical stability as well as maintain its antioxidant and anti-inflammatory activities. The produced microcapsules have been implemented into food processing and targeted drug delivery. Resistant starch is not hydrolyzed in gastric; however, it is hydrolyzed in intestinal alkaline environmental. The PC in microcapsule could successfully escape from the stomach to the intestinal action site and achieve its sustained-release ability. The PC, when combined with resistant starch, can promote the formation of a hydrogel network under their swelling state to hinder the PC release and prolong the functional time in vivo [66]. This encapsulation system can overcome the physical and chemical degradation in the gastrointestinal tract and reduce the contact between PC and gastrointestinal fluid.

Chemical self-assembly of polysaccharides has become an effective method for the microencapsulation of PC. Polymer modification technologies such as crosslinking, grafting, and substitution can improve the encapsulation efficiency by increasing the molecular weight, aggregation state, chain structure, and physicochemical properties of microcapsule wall materials to form polymer networks to capture protein molecules more easily. Studies have showed that, when the PC was coated with the modified porous starch, compared to the unmodified starch, its encapsulation efficiency and loading capacity increased by 77.27% and 135.10%, respectively. Thus, the modified porous starch microcapsules could effectively deliver PC to the gastrointestinal tract, maintain its sustained release, and achieve good fluorescence imaging in vivo [67].

Chitosan contains amino, acetylamino, and hydroxyl groups which can form a gel barrier under acidic conditions and make chitosan stable against enzymes in the duodenum and the lower intestinal tract. It is a food-grade biopolymer that can be used for the intestinal target area. Chitosan encapsulates PC, which is protected from gastric acid degradation; then, PC is released in the intestinal environment. Thus, the targeted therapy is achieved [45]. The nanoparticles formed by the encapsulation of PC with carboxymethyl chitosan could be used for anti-cancer drugs, which effectively targeted the surface of HeLa cells and significantly inhibited the proliferation of HeLa cells [47].

Protein, chosen as the wall material of microcapsules, is mainly due to its good emulsification and film-forming ability. The film-forming wall material encapsulates the core compound and then protects the active substance from being destroyed by the external environment. A few researchers reported that some proteins improved the storage stability of PC by wrapping PC chromophores in the protein networks and reduced the attack of external oxidative free radicals on PC [68]. Zhang et al. [69] found that isolated whey proteins could stabilize the color of PC under light, heat, and acid environment for 5 days, while the contrast could only exist for 3 days.

### 3.2. Synthetic Material

Protein bioactive substances show great application potential as products in the pharmaceutical and healthcare industries. PC assembled with other polymers will improve the stability of process and storage and can be used in the development in functional foods, targeted drugs, and other fields.

At present, more and more synthetic delivery vectors are being used for PC loading. The combined carrier of PEG-b-(PG-g-PEI) was made from low molecular weight polyethylene imine (PEI), polyethylene glycol (PEG), and poly-L-glutamic acid (PG). When it was used to encapsulate, its positive charge could bind PC by electrostatic interaction. Animal experiments showed that the carrier exhibited the sustained-release characteristics, increased the plasma half-life of PC in mice, reduced oxidative stress damage and islet cell apoptosis, and protected pancreatic islets functions in mice [70].

Chen et al. [71] prepared metal–organic skeleton material zeolitic imidazolate frameworks-8 (ZIF-8) by biomimetic mineralization method to encapsulate PC. The organic skeleton material ZIF-8 utilized the static interaction between PC and metal ions to form microcapsules. ZIF-8 responded to the external acid-base environment, slowly released and prolonged the action time of PC in vivo, and realized the accurate delivery of PC in cells.

Synthetic materials have better thermal and chemical stability than natural materials and are widely used. However, manufacturing or modifying some synthetic materials often involve toxic agents, such as PEI and formaldehyde [72], leading to an increase in public concern, especially when it used in products in the food and health industry.

## 4. Microencapsulation Technology of PC

### 4.1. Spray Drying

Spray drying is a technique involving water removal from the material through contact with hot air after the material is dispersed into micro-particles by the mechanical centrifugal force or a pressure difference. The microcapsule wall material along with the water is believed to wrap the thermo-sensitive PC in the core through electrostatic attraction and hydrogen bonding due to the dissociation of a large number of functional groups of wall materials and PC, such as -CH_3_, -NH_2_, -COOH, -CONH_2_, -OH, etc., [19,20]. In the spray drying process, the water of the wall material is evaporated as the heat releases and temperature increases slowly to achieve the wet-bulb temperature of the air. This process effectively protects the PC from damage during the heat treatment and demonstrates the purpose of the microencapsulation technology. The spray drying treatment can make the wall material of microcapsules form a dense barrier, which improves the encapsulation and the protection of core materials. This technology, being used for the PC microencapsulation, exhibits high yield and low processing cost. The major morphology of microcapsules produced by the spray drying is a multi-core type. Meanwhile, the microcapsules could also form a composite type if the produced microcapsules are re-encapsulated with other compounds during the spray drying. The inlet temperature and wall materials are significant factors that affect the types of microcapsules [73]. Iqbal reported that the optimum inlet temperature of dried PC microcapsules was 110 °C [21]. The technical schematic is shown in Figure 3.

The encapsulation wall materials applied in the spray drying technology are commonly soluble in water and have good thermal stability. Maltodextrin is a polyhydroxy compound, which can form a protective layer by secondary bonds such as hydrogen bonds to encapsulate proteins, reduce the collision between protein molecules, inhibit protein hydrolysis, and endow it with good thermal stability. Faieta et al. [22] used maltodextrin and trehalose to construct PC microcapsules with high carrying capacity. After the encapsulation of the microcapsules, the residual amount of PC was more than 89%, and the concentration of maltodextrin was positively correlated to the thermal degradation resistance and color protection ability of PC. Da Silva et al. [23] prepared the alginate extract microcapsules encapsulated by citric acid-crosslinked maltodextrin using the spray drying technique. The encapsulation efficiency of the microcapsules was 75%, presenting higher thermal stability than the base materials and better anti-inflammatory activity.

In the early stage of constant rate drying, the temperature around PC is not too high and does not exceed the wet-bulb temperature of the air. Along with the microcapsule powders’ drop in the bottom of the drying tower, the drying process occurs in the late stage of reduced rate drying, and the temperature in the out-layer of microcapsule the can exceed the wet-blub temperature because of a lack of excess water that is evaporated for absorbing heat. Therefore, the microcapsule contacts with the hot air again and results in the PC color change. In order to overcome this limitation, some researchers constructed a two-phase system of polyethylene glycol (PEG) and dextran to encapsulate PC. By protecting the dispersed phase of enriched PC from the continuous phase, through spray drying with continuous mixing (to prevent phase separation), double-layer enclosed microcapsules were formed. The microcapsules prepared in this system could be stored for 6 months without a change in the color of PC, showing that its stability was higher than any other malt dextrin single encapsulation systems [24]. Microcapsules of PC prepared by spray drying with maltodextrin or other water-soluble wall materials are easily soluble in water and can be used to improve the storage stability without sustained-release ability.

### 4.2. Electrospinning Technology

Electrospinning is a technique normally used for the manufacturing of polymer fibers by applying an electrostatic field force. Typically, a solution composited of core and wall material forms a polymer when exposed to the electric field through the nozzle. The produced polymer is then transferred onto the electrode and, due to the charge repulsion, is extended into fiber morphology and coated on the core material. Compared to the traditional spay drying technique, which always causes a partial loss of biological activity PC under the processing environmental conditions, the electrospinning treatment with strong electric field extrusion is usually carried out at room temperature to avoid heat denaturation and PC inactivation. The technical schematic is shown in Figure 4.

Poly(ethylene oxide) (PEO) is a water-soluble polymer material and suitable for the electrospinning nanofibers production. By mixing the PEO with PC, the PEO nanofibers produced through the electrospinning technique could expand to an average diameter of 295 nm and protected the inside PC to perform good thermal stability [25]. Moreira et al. [26] reported that an antioxidant ultrafine composite fiber with PC and PEO was made by free surface electrospinning technology. The composite fiber had a PC encapsulation rate of 5–10% (*w*/*w*) and was 269–542 nm in diameter, exhibiting a promising application in the food preservation industry. Wen et al. [27] constructed a safe, nontoxic, and probiotic-loaded PC nanofiber membrane by ion-coagulation combined with coaxial electrospinning technology. The diameter and release rate of the composite was 590 nm and 82%, respectively, and the release amount of PC in simulated colon fluids followed in a dose-time-dependent manner due to the existence of prebiotics and polysaccharides. Furthermore, the composite inhibited the growth of human colon cancer cells by blocking the cell cycle at the G0/G1 phase and inducing cell apoptosis. The mechanism involved a decrease in B-cell lymphoma-2 (Bcl-2)/ Bcl-2-associated x protein (Bax), the activation of Caspase 3, and the release of cytochrome c [28].

At the same time, it promoted the proliferation of intestinal probiotics, maintained the biological activity of PC, and realized its colon targeting. In the food quality detection field, PC-based composite nanofibers prepared by electrospinning could be used as a pH indicator to detect perishable products [29].

### 4.3. Electrospraying Technology

Electrospraying is a method that, when used under a high voltage electric field, the wall material mixture solution through the nozzle can be cracked into a charged spray, which is sprayed onto the core material surface to form a core-shell composite.

The difference between electrospraying and electrospinning is that, under the applied voltage, the electrospraying method makes polymer into ultrafine particles, while the electrospinning technology produces polymer fibers. Electrospraying technology is employed for the encapsulation of small molecular substances or proteins exhibiting good efficiency of encapsulation and high efficiency of producing monodisperse particles. The rapid encapsulation of compounds can be realized by electrical spraying technology to prepare bioactive microspheres with smooth surfaces and small particle sizes [30]. Several researchers used polyvinyl alcohol (PVA) ultrafine particles to spray onto PC, showing 75% encapsulation efficiency and producing the compound with 395 nm as the average diameter. The microcapsules could tolerate the temperature up to 216 °C and still maintained the antioxidant activity of PC [31]. Figure 5 shows the preparation process of PC microcapsules by electrospraying.

Microcapsules encapsulated by electrospraying have specific particle states and dispersion properties and keep the biological activity unchanged [32]. However, higher requirements are needed regarding the viscosity, fluidity, and constituent and solvent volatility of the solution system to meet the electrospraying technique. The irregular movement of charged particles during spraying results in a loss of wall material and an uneven distribution of the coating.

### 4.4. Liposome Delivery

Liposomes are small vesicles formed by lipids or a liposoluble substance, such as phospholipids or cholesterol, which is dissolved in organic solvents and then added to water. Liposomes have a bimolecular layer structure similar to the cell membrane and can encapsulate hydrophilic substances in a water cavity acting as the controlled-release substances [33]. The microcapsule formed from liposomes encapsulating PC is shown in Figure 6. This microcapsule can effectively improve the stability and protect the protein compounds’ oral bioavailability; therefore, they can smoothly pass through the acidic environment of the stomach without any damage, thereby maintaining their anti-inflammatory activity [34].

Yagoubi et al. [35] used an ultrasound-assisted method to prepare solid lipid-nanocarriers for microencapsulated PC producing a uniform spherical particle with a particle size of 50–80 nm. The encapsulation efficiency of this process is 37–69%. These lipid nanocarriers can control the release of PC.

Hardiningtyas et al. [36] prepared composite particles with the particle size of 220–270 nm by forming water in oil (W/O) emulsion with more than 99% encapsulation efficiency. The microencapsulated PC effectively maintained the active structure of PC and released it. Castangia et al. [37] used sodium hyaluronate to immobilize liposomes prepared by concentrated soybean phosphatidylcholine, and the obtained multilayer liposomes had a unique hyaluronic acid–phospholipid structure, which was conducive to the maintenance of PC activity and deep accumulation. Liposomes can be used as osmotic carriers to encapsulate PC for targeted delivery; however, the yield of water-soluble compounds encapsulated by liposomes is low, and the microcapsules produced are easily oxidized in the air, which greatly limits their applications.

### 4.5. Sharp-Hole Coagulation Bath

The sharp-hole coagulation bath method is a technique that involves dispersing the core material in a soluble polymer solution and adding spherical droplets into the coagulation bath with the help of the sharp-hole instrument for rapid precipitation or crosslinking solidification to form the cystic structure complex. Sodium alginate presents good biological activity and biocompatibility [38], and it is commonly used as condensation material. With the negative charge, sodium alginate can form gel by complexing with calcium ions, which was further employed to encapsulate PC to form calcium alginate/PC microcapsules through extrusion [39], as shown in Figure 7. Moreover, the encapsulation efficiency of microcapsules prepared by this method can reach 98%.

The calcium alginate/PC microcapsules treated by the sharp-hole coagulation bath were spherical, with an average size of about 1.2 mm. The addition of calcium alginate did not destroy the structure of PC. Moreover, the produced compound could tolerate an acidic environment with a pH of 4.5 and demonstrates a degradation rate that is slower than unencapsulated PC [40,41]. The sharp-hole coagulation bath method is also suitable for the complex system of alginate and other substances. Yan et al. [42] used alginate and chitosan as coating materials to prepare PC microcapsules by the extrusion method, forming a dense spherical complex with an average diameter of 1.03 mm. The encapsulation of alginate and chitosan can reduce the temperature sensitivity of PC during storage and can enhance the sustained-release effect of single alginate-encapsulated microcapsules due to the addition of chitosan.

### 4.6. Ion Gelation

Ion gelation is a method used to encapsulate bioactive substances by which electrolyte materials are normally electrolyzed into anions and cations during the polymer molecular chains’ crosslink to form a spatial network structure through electrostatic attraction or forming hydrogen bonds. The ion gelation method is a new preparation method for drug-loaded microcapsules with mild preparation conditions. In the field of healthcare products and medicine, ion gelation can use PC composite photosensitizers to assist in antibacterial photodynamic therapy [43].

Mohan et al. [44] prepared chitosan nanoparticles with an average particle size of 20 nm by using the ion gelation method, exhibiting 36.45% encapsulation efficiency of PC and maintaining the antibacterial activity of PC. The negative-charged sodium tripolyphosphate could combine with the positive-charged chitosan by ion gelation method to encapsulate the anti-tumor polypeptide Y2 from hydrolyzed PC to form microcapsules. The maximum encapsulation efficiency was 49% with 15% polypeptide content [45]. In addition, natural food additives, carrageenan, and PC can also form porous network materials with hydrophilic and mechanical strength through ion crosslinking [46]. To realize the oral administration of PC, Yang et al. [47] used carboxymethyl chitosan to prepare PC complexes by ion gel method. The particles were spherical, with a diameter of about 200 nm, and with 65% encapsulation efficiency. The nanoparticles were pH sensitive to the release of PC and had sustained-release effect in vitro. Manconi et al. [48], utilizing chitosan and xanthan gum as polyelectrolyte complexes to encapsulate PC (shown in Figure 8), found that, when the mass ratio of chitosan and xanthan gum was 0.5/8.0 (*w*/*w*), the surface of the microcapsules, after drying, was regular. The microcapsules satisfy the Fick diffusion characteristics of drug release and presented good in vitro adhesion, which could be useful in colon administration.

## 5. Prospect of PC Microencapsulation

In recent years, increasing evidence has demonstrated PC’s bioactive functions and action mechanisms; however, its poor chemical stability and low bioavailability limit its applications in many fields, especially in food and medicine. Embedding technologies, such as microencapsulated technologies, can remarkably improve PC chemical stability; thus, they successfully help PC resist adversity, finishing the target delivery and expanding the application field. However, there are some shortcomings in developing the PC microencapsulated technologies to be overcome, such as the improvement of the preparation process for nonuse toxic reagents, utilizing safe raw materials, and increasing bioavailability for oral delivery. Therefore, future PC encapsulation technologies should employ a safe preparation process and present a high PC encapsulation rate [74,75]. In addition, with the improved manufacturing techniques and the development of new approaches for site-specific vector targeting, encapsulated PC is believed to play an important role in increasing the efficacy of functional foods and even drugs in the future.

## Figures and Tables

**Figure 1 molecules-27-05854-f001:**
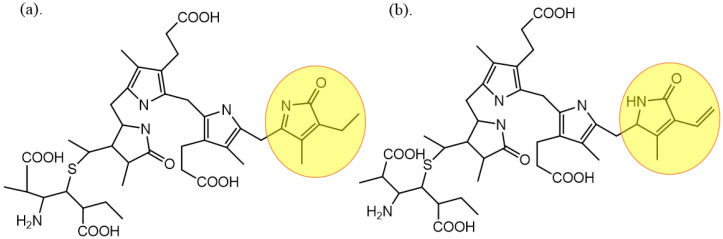
The structure of PCB (**a**) and PEB (**b**) (the structural differences are highlighted).

**Figure 2 molecules-27-05854-f002:**
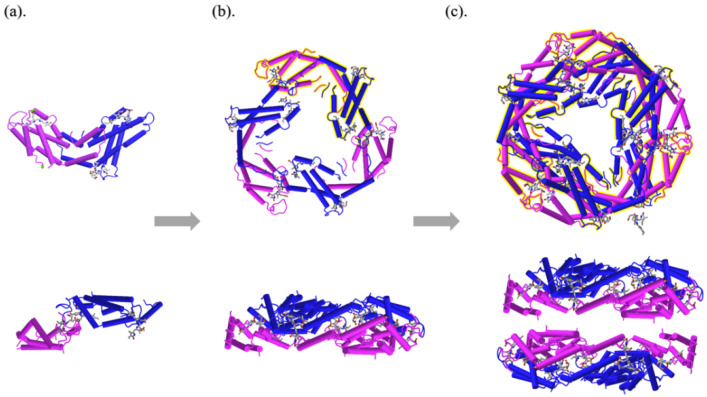
Cylinder and plate representation of PC monomer (**a**), PC trimer (**b**), and PC hexamer (**c**).

**Figure 3 molecules-27-05854-f003:**
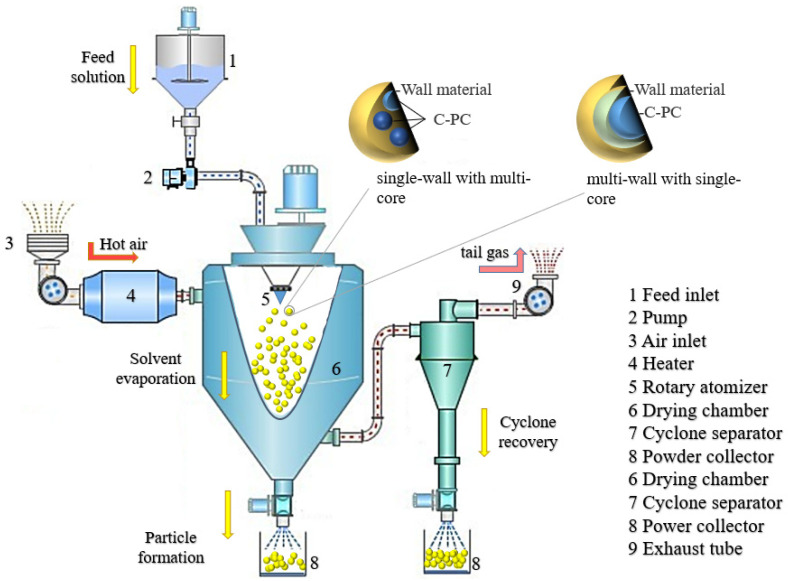
Schematic illustration of the process for preparing PC microcapsules through the spray drying technique.

**Figure 4 molecules-27-05854-f004:**
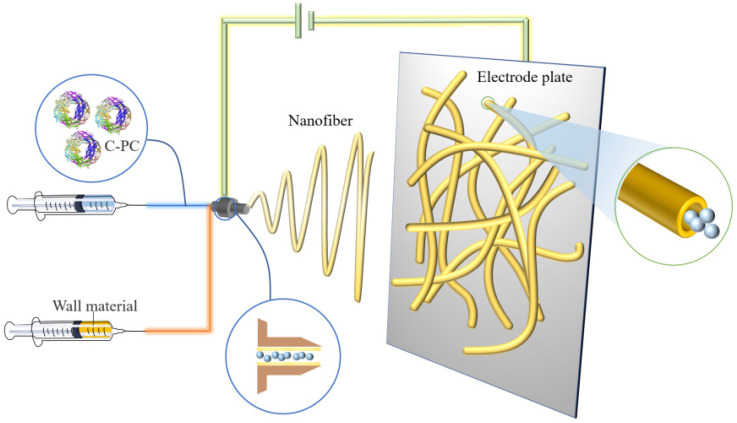
Schematic illustration of preparing PC microcapsules using the electrospinning method.

**Figure 5 molecules-27-05854-f005:**
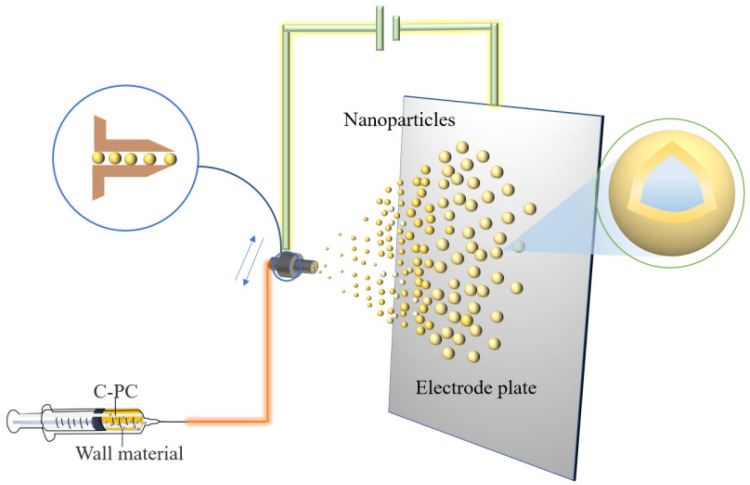
Schematic illustration showing the preparation of PC microcapsules using the electrospraying method.

**Figure 6 molecules-27-05854-f006:**
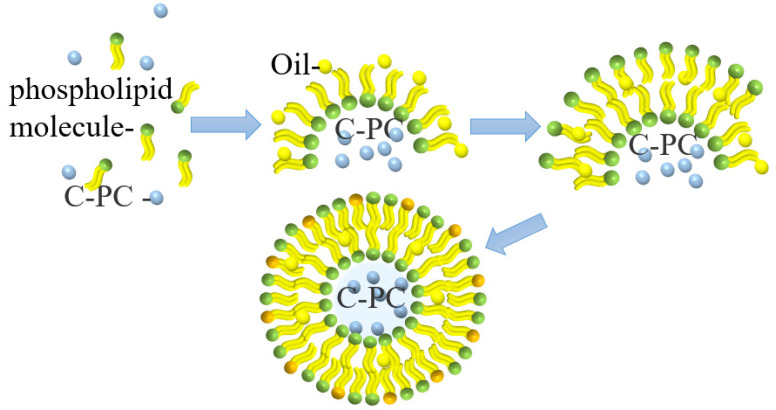
Schematic illustration of PC liposome microcapsule formation.

**Figure 7 molecules-27-05854-f007:**
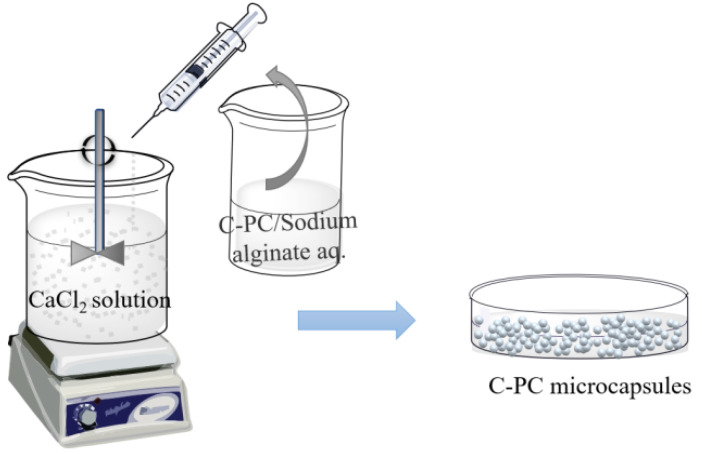
Schematic illustration showing the preparation of PC microcapsules using the sharp-hole coagulation bath method.

**Figure 8 molecules-27-05854-f008:**
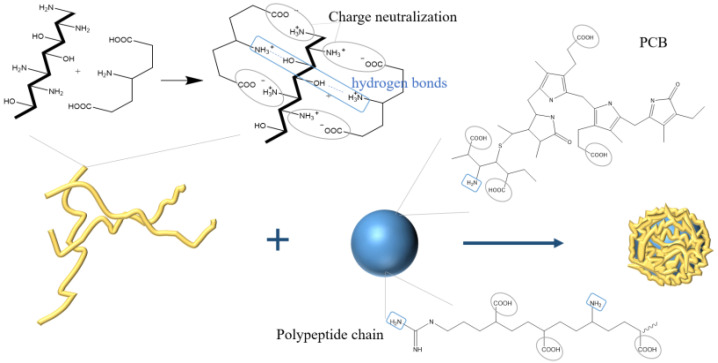
Schematic illustration showing the preparation of PC microcapsules using ion gelation method.

**Table 1 molecules-27-05854-t001:** Overview of the advantages and disadvantages of various methods for preparing PC microcapsules.

Method	Benefits	Drawbacks	Reference
Spray drying	Rapid, cost-effective, easy to scale up. C-PC retention, and relatively good storage stability.	Production of nonuniform particles with a wide size distribution. C-PC degradation and loss of product.	[19,20,21,22,23,24]
Electrospinning	Suitable for heat-sensitive compounds.	Slow, produces low encapsulation yield, and limited scope of application.	[25,26,27,28,29]
Electrospraying	Produces particles with a high surface-to-volume ratio, controlled release, improved functionality, and physical properties.	Time-consuming and hardly repeatable, especially at the industrial level.	[30,31,32]
Liposome delivery	Increased adsorption bioavailability, and nontoxic.	Low encapsulation efficiency. Lipid oxidation, poor physicochemical stability, and wide particle size distribution. Postprocess step is required.	[33,34,35,36,37]
Sharp-hole coagulation bath	High loading capacity, low temperature operating requirements, reduced evaporation losses of volatile compounds, and thermal degradation. Tailored release of active compounds.	Special instruments are required and are not suitable for large-scale industrial production.	[38,39,40,41,42]
Ion Gelation	Low cost and does not require advanced equipment, high temperatures, and organic solvents.	Produced particles are very sensitive to pH and ionic strength. Agglomeration of particles and particle size control.	[43,44,45,46,47,48,49]

## Data Availability

Not applicable.

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
