# Peer review of "Progress of Microencapsulated Phycocyanin in Food and Pharma Industries: A Review"

_molecules, 2022, doi:10.3390/molecules27185854_

Round 1

Reviewer 1 Report

This review highlights the significance of the microcapsules’ wall materials which commonly protected phycocyanin from environmental interference, and summarizes the current preparation principles and characteristics of microcapsules in food and pharma industries. Moreover, the major technical challenge and corresponding countermeasures of phycocyanin microencapsulation are also appraised, providing insights for the phycocyanin’ broader application. However, there are several minor concerns that need to be addressed before publication.

1.        The author should check the typos and grammar more carefully, such as line 153 “amylase” or “amylose”? Line 170 “lower intestine”? Line 182 “…, it’s encapsulation efficiency and …” this sentence should be revised.

2.        All abbreviations should be given at the first time, and then just directly use them, such as TLR-4/ Myd 88/ NF-κB, ZIF-8,

3.        The keywords should be changed, such as pigment.

4.        The figure 1 and 2 should be highlighted the different of structure of compounds.

5.        Section 3.1 Line 130-144, I think this paragraph should be placed before title 3.1 as a generalization.

6.        Line 176-185, I think this paragraph should be placed in starch section, because modified porous starch is also a form of starch.

7.        Line 193 ”high level” should be given the values according to the report of previous study.

8.        Figure 3 the schematic illustration of the spray drying should be changed, please refer to the other references (Title: Trends of spray drying: A critical review on drying of fruit and vegetable juices).

9.        Section 5. I don't think it's necessary to repeat the previous content, just simply and directly explain the development direction of PC microcapsules or the possible research strength.

10.     The formation of references should be revised according to the journal of “guide for authors”.

Author Response

Dear reviewer:

We would like to express our sincere gratitude to you for your thoughtful, constructive, and positive feedback, which helped us improve the paper! Below are responses to your comments. Please see the attachment.

Your continuous suggestion and comments on our revised manuscript will be greatly appreciated. We are looking forward to your further comments.

Sincerely yours,

Xueqing Wang

Reviewer 2 Report

- The topic is interesting, the manuscript of general interest treat a subject quite well documented by the authors. The subject of current investigation and method carried out are within the scope of the Molecules Journal.

However, a major revision is required before the paper can be considered for publication as follows:

- Generally, the English grammar is poor, several words have incorrect spelling, so the writing style needs to be improved. A scan by a native English speaker or a Professional English Language Editing Service is advisable in order to avoid " Starch is polysaccharide polymers", " have been implemented into in food", " starch is resistant from hydrolysis by gastric acid", etc.

- Authors are asked to define the term " decoproteins"

- I don't understand the meaning of " and forms a protein-chromophore complex, like Spirulina platensis, by utilizing which, for photosynthesis". Authors are asked to rewrite the phrase.

- page 5: authors mention that starch used for encapsulation improve the PC stability. The term "stability" is too broad? Authors are asked to be more precise (thermal, chemical stability?).

- page 6: Cite at least one reference to support  "The microcapsule wall material along with the water is believed to wrap the thermosensitive PC in the core through electrostatic attraction and hydrogen bond. "

- the references cited in page 2 (Table 1) are not in order (reference 16 in text followed by reference 65 in Table 1 in the same page).

- the quality of Figure 3 is poor.

Author Response

(The authors gave the same response as above.)

Reviewer 3 Report

This review is on delivery systems for a specific compound, Phycocyanobilin, there is no justification while this compound is selected for a full review. The authors describe this molecule as a polypeptide, it is not. 

There is little interest on delivery systems for a specific compound that its importance was not specified. 

Author Response

Dear reviewer:

We would like to express our sincere gratitude to you for your thoughtful, constructive, and positive feedback, which helped us improve the paper! Below are responses to your comments. Hopefully we have addressed all your concerns. Please see the attachment.

Your continuous suggestion and comments on our revised manuscript will be greatly appreciated. We are looking forward to your further comments.

Sincerely yours,

Xueqing Wang

Round 2

Reviewer 2 Report

Manuscript accepted in the present form.

Reviewer 3 Report

the revised article is suited for publication